# Effects of Pretreatment and Polarization Shielding on EK-PRB of Fe/Mn/C-LDH for Remediation of Arsenic Contaminated Soils

**DOI:** 10.3390/nano13020325

**Published:** 2023-01-12

**Authors:** Zongqiang Zhu, Yusong Kong, Hongqu Yang, Yan Tian, Xiaobin Zhou, Yinian Zhu, Zhanqiang Fang, Lihao Zhang, Shen Tang, Yinming Fan

**Affiliations:** 1Collaborative Innovation Center for Water Pollution Control and Water Safety in Karst Area, Guilin University of Technology, Guilin 541004, China; 2The Guangxi Key Laboratory of Theory and Technology for Environmental Pollution Control, Guilin University of Technology, Guilin 541004, China; 3Technical Innovation Center of Mine Geological Environmental Restoration Engineering in Southern Karst Area, Nanning 530022, China; 4Chongqing Hechuan Ecology and Environment Monitoring Station, Chongqing 401519, China; 5School of Chemistry and Environment, South China Normal University, Guangzhou 510006, China

**Keywords:** pretreatment, polarization shielding, arsenic-contaminated soil, electrokinetic remediation, layered double hydroxide

## Abstract

In this study, coupling electrokinetic (EK) with the permeable reactive barriers (PRB) of Fe/Mn/C-LDH composite was applied for the remediation of arsenic-contaminated soils. By using self-made Fe/Mn/C-LDH materials as PRB filler, the effects of pretreatment and polarization shielding on EK-PRB of Fe/Mn/C-LDH for remediation of arsenic contaminated soils were investigated. For the pretreatment, phosphoric acid, phosphoric acid and water washing, and phosphate were adopted to reduce the influence of iron in soil. The addition of phosphate could effectively reduce the soil leaching toxicity concentration. The removal rate of the soil pretreated with phosphoric acid or phosphoric acid and water washing was better than with phosphate pretreatment. For the polarization shielding, circulating electrolyte, electrolyte type, anion and cation membranes, and the exchange of cathode and anode were investigated. The electrolyte circulates from the cathode chamber to the anode chamber through the peristaltic pump to control the pH value of the electrolyte, and the highest arsenic toxicity removal rate in the soil reaches 97.36%. The variation of total arsenic residue in soil using anion and cation membranes is the most regular. The total arsenic residue gradually decreases from cathode to anode. Electrode exchange can neutralize H^+^ and OH^−^ produced by electrolyte, reduce the accumulation of soil cathode area, shield the reduction of repair efficiency caused by resistance polarization, enhance current, and improve the removal rate of arsenic in soil.

## 1. Introduction

Soil arsenic (As) pollution has attracted increased attention due to its strong carcinogenicity and high toxicity threat to human health and food safety [1,2]. In situ immobilization treatment is a widely used remediation method for As-contaminated soil as it has the advantages of being non-invasive, fast and effective [3,4]. For in situ immobilization, electrokinetic remediation (EK) is a promising technique, which can eliminate multiple heavy metals simultaneously [5,6]. However, EK technology only collects pollutants in electrode chambers or transfers them to one side of the soil, then the pollutants still need secondary treatment [7]. Research has shown that EK technology combined with permeable reactive barriers (PRB) can overcome this difficulty, by migrating arsenic to the reactive materials in PRB using electrodialysis, electrophoresis, electromigration or other mechanisms [8,9]. EK-PRB technology was used to treat soil polluted by tetrachloroethylene using zero-valent iron as the filler. Under the best experimental conditions, the removal rate of tetrachloroethylene reached 80% [10]. Remediation of chromium (VI) pollution in water using in situ electrodynamic force and a permeable reactive wall produced a total chromium removal rate reaching more than 90% [11]. At the same time, EK-PRB technology improved the removal rate of arsenic in soil from 42% to 57% using EK alone [12]. The reducible arsenic that is difficult to remove from the soil is transformed into acid soluble arsenic that is easy to remove. In addition to remediation of heavy metal contaminated soil, it can also repair organic contaminated soil and soil contaminated by low permeability pollutants. This technology is not only relatively unaffected by external conditions, but also avoids adding any environmentally unfriendly substances in the remediation process. It is a low cost and green remediation technology for contaminated soil [13].

The efficiency of heavy metal removal depends on PRB materials used, such as activated carbon, layered double hydroxide (LDH) and ion exchange membrane [14]. Taking advantage of the high capacity for anionic exchanging, LDH is a promising PRB material [15]. It is often applied as an adsorption material, known as “anionic clay”, which has a wide range of compositions [16]. However, arsenic has an intensive affinity to Mn, Fe and Al oxides in soils, leading to low removal rate and poor mobility [17,18]. Therefore, investigate the main effect factors for improving As-contaminated soil remediation with EK-PRB is necessary. In actual soil remediation, the electric drive enhancement process is greatly affected by the physical and chemical properties and iron content in red soil. In order to clarify the impact of soil iron content and its physical and chemical properties on arsenic pollution remediation, different pretreatment methods and polarization shielding were designed to reduce the impact of soil iron on electric drive adsorption and clarify its basic application principle. For example, controlling the pH value of the cathode and anode within a certain range can effectively improve the remediation effect of contaminated soil [19]. The addition of acid electrolyte citric acid to the cathode can effectively control the pH value of the cathode, and the addition of alkaline electrolyte to the anode can neutralize the H^+^ produced by the anode, so as to improve the effect of soil remediation [20,21].

Due to layered double hydroxides and carbon nanomaterials (C-LDHs) having versatile properties and many composition variables available for tuning, hybrids of them are promising nanomaterials. In this work, Fe/Mn/C-LDH materials made as PRB filler, the effects of pretreatment and polarization shielding on EK-PRB for remediation of arsenic contaminated soils (including iron content, phosphoric acid, phosphoric acid and water washing, phosphate, circulating electrolytes, electrolyte type, anion and cation membranes), and the exchange of cathode and anode were investigated.

## 2. Materials and Methods

### 2.1. Materials

#### 2.1.1. Soil

The experiment soil was acquired from Hechi City, Guangxi Zhuang Autonomous Region, China. Detailed pretreatment steps of the soil are included in the Appendix A; physical and chemical properties of soil before and after pollution are shown in Appendix A.

#### 2.1.2. Preparation of PRB Filler Fe/Mn/C-LDH

The Fe/Mn/C-LDH material was made from bamboo. First, bamboo chips were boiled at 100 °C in 5% dilute ammonia water for 6 h, then dried at 80 °C for 24 h, and roasted at 600 °C in muffle furnaces for 3 h. Afterwards, the bamboo charcoals were soaked at 110 °C in concentrated HNO_3_ for 2 h, then washed with deionized water and dried at 80 °C to acquire the bamboo charcoal bio-templates. The bamboo charcoal bio-templates were added into deionized water, 1 M Fe and Mn solution was added (Fe/Mn molar ratio = 2:1), and the pH was adjusted to 11.5 by adding a solution of 0.1 M Na_2_CO_3_ and 3.2 M NaOH. Finally, the solution was stirred for 2 h and dried to acquire Fe/Mn/C-LDH materials. The chemicals were of analytical grade or higher and purchased from the Shanghai Guoyao reagent group, Shanghai, China.

### 2.2. Construction and Operation of EK-PRB Device

The experiment was carried out in an EK-PRB repair device, which is shown in Figure 1. The device was mainly composed of a polymethyl methacrylate chamber (300 mm × 100 mm × 50 mm), including electrode, direct current power supply and PRB filler. The anode and cathode were graphite plates (100 mm × 10 mm × 50 mm), and the Fe/Mn/C-LDH material was used as the PRB filler. In each run for 96 h, 600 g of the arsenic-contaminated soil was placed into the soil chambers. The voltage, PRB location, moisture content, PRB types and pretreatment of arsenic-contaminated soils were varied to investigate the effects of the operation condition. The detailed composition of the EK-PRB device is included in the Appendix A. The physical diagram is shown in Appendix A.

### 2.3. Analysis Methods

#### 2.3.1. Soil Physical and Chemical Properties

After each run, the treated soil was sectionally sampled, dried, ground and sieved through a 100-mesh sieve. The detailed steps are included in the Appendix A.

#### 2.3.2. Analysis Method for Arsenic Content and Arsenic Forms

The leachability of arsenic in the treated soils was tested by employing the in vitro toxicity characteristic leaching procedure (TCLP) [22]. The detail steps are included in the Appendix A. A modified four-step extraction procedure (BCR, European Community Bureau of Reference) was applied to detect the chemical speciation of arsenic in soils [23]. The detailed steps are included in Appendix A.

## 3. Results and Discussion

### 3.1. The Effect of Pretreatment on Arsenic Pollution Remediation

#### 3.1.1. Phosphoric Acid

The presence of iron has a certain effect on the migration of electrically driven arsenic removal. Therefore, reducing the content of original iron in soil is very important for EK-PRB to repair arsenic contaminated soil. Acid washing could improve the mobility of arsenic [24]. Thus, a pretreatment method of phosphoric acid was designed to reduce the effect of iron in soil on electrical drive. As shown in Figure 2a, after the pretreatment of phosphoric acid, the concentrations of residual arsenic in soil under EK remediation was 213.91 mg/kg, and the removal rate of arsenic was 57.21%. Under EK-PRB remediation, the concentration of residual arsenic was 258.66 mg/kg, and the removal rate of arsenic was 48.27%, which increased by about two times compared to absence of pretreatment. It was found that EK and EK-PRB were effective in the remediation of arsenic contaminated soil pretreatment with phosphoric acid. Figure 2b showed after being pretreated with phosphoric acid, the removal rates of EK and EK-PRB were 70.60% and 99.20%, respectively. Due to the high leaching toxicity of arsenic contaminated soil pretreated with phosphoric acid and the limited purification capacity of PRB material, the remediation advantage of pretreatment combined with EK-PRB was not obvious in terms of removal rate. From the perspective of leaching toxicity content and leaching toxicity removal, the leaching toxicity of soil pretreated with phosphoric acid reached 213.2 mg/kg, which was two times higher than that without pretreated. Besides, the leaching toxicity concentration after EK remediation reached 62.65 mg/kg. By comparison, the leaching toxicity concentration after EK-PRB remediation reached 1.97 mg/kg. It was found the effect of phosphoric acid on the leaching toxicity concentration of soil was better than the effect of the uneven distribution of soil pH caused by EK and the solidification and stabilization of soil by Fe ions flowing out during the action of PRB material.

The variations of current density as a function of time after the pretreatment of phosphoric acid is shown in Figure 2c. Although the change trend of current increased first, then decreased and finally tended to be stable, the current density after the pretreatment of phosphoric acid was obviously higher than that without pretreatment. The reason might be that the ions in soil increased after the pretreatment of phosphoric acid, which enhanced the electric driving ability and increased the current density. The effect of phosphoric acid pretreatment on the soil pH value of the system is shown in Figure 2d. The change trend of soil pH was gradually increasing from anode to cathode, but the overall soil pH value of arsenic contaminated soil pretreated with phosphoric acid was lower than without pretreatment, which was related to the initial pH value of soil after phosphoric acid pretreatment. In addition, the soil pH value of EK-PRB remediation process was slightly higher than that of EK remediation process. The reason might be that Fe/Mn/C-LDH was alkaline material neutralized part of the acidity of the surrounding soil.

Figure 2e shows the distribution of EC after phosphoric acid pretreatment. After EK and EK-PRB remediation, the EC at anode increased from 151.30 to 2236.50 and 2426.50 μS/cm, respectively. The EC of soil without pretreatment increased to 3001.00 μS/cm. It was found that phosphoric acid pretreatment changed the soil pH and indirectly affected EC. Phosphoric acid pretreatment of soil would affect the pH value of soil, thus affecting the relevant parameters of the soil remediation process. Considering that the addition of phosphoric acid would improve the removal effect of arsenic, the influence and the mechanism of its electric driving process were unknown. Therefore, a comparative experimental group of phosphoric acid and water washing was designed to remove the interference of phosphoric acid pretreatment on free arsenic ions and other influencing factors in soil. This was more conducive to clarifying the process of EK remediation and EK-PRB remediation after phosphoric acid pretreatment.

#### 3.1.2. Phosphoric Acid and Water Washing

The change of soil arsenic residue after phosphoric acid and water washing pretreatment is shown in Figure 3a. The arsenic residues of EK and EK-PRB pretreated with phosphoric acid and water washing were 261.20 and 249.59 mg/kg, respectively. Compared with non-pretreatment, the arsenic residue of EK-PRB improved 26% and only 7% less than phosphoric acid pretreatment. The removal effect of arsenic with phosphoric acid and water washing pretreatment was slightly less than phosphoric acid pretreatment and much higher than without pretreatment. Figure 3b illustrates that the leaching toxicity removal rate of arsenic contaminated soil was much higher than that of phosphoric acid pretreatment, which was still lower than without pretreatment. Moreover, the leaching toxicity removal rate of soil pretreated with phosphoric acid and water washing by EK-PRB remediation was uneven. Specifically, the lowest removal rate was only 50% near the cathode, while it was 96.8% at the anode. The reason might be that Fe/Mn/C-LDH had a passivation effect on target pollutants, which increased the removal rate [25].

In Figure 3c, the change trend of current increased within 48 h, then decreased and finally tended to be stable. The reason is that the current would increase rapidly since H^+^, K^+^, Cl^−^, and OH^−^ under the action of the electric field would move to the cathode and anode, respectively. However, with the increase of treatment time, the concentration of mobile ions in soil decreased, resulting in the decrease of current density. Figure 3d shows the change of pH in soil. The change of pH was due to H^+^ and OH^−^ produced by the electrolysis of electrolyte made the anode acidic and the cathode alkaline. H^+^ and OH^−^ would move to the anode and cathode, respectively, so that the pH of the soil gradually rose from the anode to the cathode. In addition, the change slope of soil pH was greater after phosphoric acid and water washing pretreatment. The reason might be that the soil structure after phosphoric acid and water washing pretreatment was looser and the trend of ion movement was more obvious.

As shown in Figure 3e, there was no obvious change trend in EC of arsenic-contaminated soil pretreated with phosphoric acid and water washing. Since the pH value of the soil near the anode was extremely low, it was conducive to the dissolution of insoluble ions in the soil, which increased the EC of soil through improving the number of ions in soil. However, the decrease of EC in soil was because there was a large amount of OH^−^ at the cathode, which would combine with other metal ions to form a precipitate. In addition, by comparison, the EC at the cathode was higher than the distance from the cathode, which was 2.5 cm. The reason was that the soil near the cathode would be in direct contact with the electrolyte and contain more ions, resulting in the increase of EC.

#### 3.1.3. Phosphate

Although phosphoric acid pretreatment could effectively increase the removal effect of arsenic, it would change the original pH value and the physical and chemical properties of the soil. Zhang et al. (2010) found that adding a chemical regent could improve the mobility of arsenic [26]. In addition, enhanced As leaching upon phosphate addition was reported by Theodoratos et al. (2002) [27] and Basta and McGowen (2004) [28]. Therefore, phosphate (KH_2_PO_4_) combined with EK-PRB was designed to explore its removal effect and repair mechanism on arsenic-contaminated soil. The distribution of soil arsenic residue after phosphate pretreatment is shown in Figure 4a. The removal effect of arsenic after phosphate pretreatment was lower than phosphoric acid pretreatment or phosphoric acid and water washing pretreatment. In addition, the concentration of arsenic residue in the middle part was higher than that in the original soil. The reason might be that the addition of phosphate made arsenic form precipitation and solidify stably in soil, which resulted in poor migration ability. In the middle part, the concentration of arsenic residue after EK-PRB remediation was lower than that after EK remediation. The reason was that PRB filler had a certain adsorption and removal effect on arsenic. In Figure 4b, the leaching toxicity removal rate of soil pretreated with phosphate is obviously higher than that without pretreatment and was also higher than that pretreated with phosphoric acid or phosphoric acid and water washing. The reason might be that the addition of phosphate reduced the mobility of arsenic and made arsenic passivated in soil, resulting in reducing the leaching toxicity concentration and harm to organisms and ecology.

As shown in Figure 4c, the current density of the experiments after EK-PRB remediation was lower than EK remediation. It was considered that the resistance of EK-PRB system increased due to the existence of PRB, which reduced the current density. After 48 h, the current density decreased gradually, which might be related to electrode polarization. Figure 4d illustrates that the overall pH of the soil with phosphate pretreatment was slightly lower than without pretreatment. The reason might be that H_2_PO_4_^−^ could resolve OH^−^ from the soil colloid through ion exchange to improve the soil pH. However, the concentration of K^+^ in soil increased due to the addition of phosphate, and its mobility in soil was stronger than H_2_PO_4_^−^. On the soil colloid, a large number of H^+^ could be preferentially resolved through ion exchange, so as to inhibit the release of OH^−^ and reduce the soil pH. Figure 4e shows a similar trend of EC with pH in soil. The EC of soil pretreated with phosphate was higher than that without pretreatment in the middle part. This occurred because phosphate might interact with the soil and release more alkaloids and salts, resulting in a significant increase in EC [29]. It also might be that phosphate made arsenic passivate and then moved to the middle of the soil through electric drive. The dissolution of some ions from PRB filler made the EC of soil in the middle part higher than the original soil.

### 3.2. The Effect of Polarization Shielding on Arsenic Pollution Remediation

#### 3.2.1. Circulating Electrolyte

As shown in Figure 5a, the residual amount of arsenic in soil is the largest at a distance of 2.5 cm from the cathode. It may be that the migration speed of OH^−^ generated at the cathode is slower than that of H^+^ generated at the anode [30], causing heavy metal ions near the cathode to form precipitation in this area, thus affecting the migration of arsenic. The residual amount of arsenic in the soil 5 cm from the cathode (i.e., the middle position) is slightly lower, which may be due to the adsorption of PRB materials in the middle position. In normal soil, arsenic exists in the form of oxygen anion in the soil, such as H_2_AsO_4_^−^, HAsO_4_^2−^ or AsO_4_^3−^. In electrokinetic repair, arsenate is transferred from the cathode to the anode because it is an oxygen anion. Therefore, the total arsenic residue in the two groups of soil at the anode is the lowest, the lowest value of circulating electrolyte is 319.00 mg/kg, and the lowest value of non-circulating electrolyte is 308.00 mg/kg. It can be seen that the circulating electrolyte has little influence on the residual amount of total arsenic in the process of EK-PRB repair technology. As shown in Figure 5b–d, in a typical EK-PRB, the H^+^ generated by anodic electrolytic reaction moves from the anode to the cathode. Accordingly, OH^−^ produced by cathodic electrolysis reaction moves from cathode to anode. Therefore, the anode area is acidic, and the cathode area is alkaline [31]. After the electrolyte circulates through the peristaltic pump, the OH^−^ generated by the cathode is removed from the cathode chamber, and the pH value of the anode chamber is increased. At the same time, the H^+^ continuously generated on the anode automatically moves to the cathode chamber through the hydraulic gradient, reducing the pH value of the cathode chamber [32]. Electric conductivity is usually defined as the reciprocal of resistivity. In the EK-PRB process, it is a comprehensive indicator reflecting the concentration and migration rate of movable ions in a pore solution. The lowest electric conductivity of the circulating and non-circulating electrolyte is 401.5 μS/cm, 404.3 μS/cm at a distance of 2.5 cm from the cathode. It starts to rise afterwards because the pH near the anode is small, and the soil is prone to acidification, which increases the soluble substances at the anode [33]. The soil conductivity near the anode is high. The circulating electrolyte can neutralize the OH^−^ and H^+^ produced by electrolysis of the cathode and anode, so that the electric conductivity on both sides of the electrode is lower than that of the non-circulating electrode solution. The leaching toxicity characteristics of arsenic in the soil after remediation are also important indicators reflecting the final remediation efficiency of contaminated soil [34]. At 7.5 cm from the cathode, the arsenic toxicity removal rate of circulating electrolyte is lower than that of the non-circulating electrolyte, and the arsenic toxicity removal rate of other points is higher than that of the non-circulating electrolyte. When the electrolyte does not circulate, the arsenic toxicity removal rate in the middle area of the soil is low, reaching the lowest value of 46.10% at 2.5 cm from the cathode, and the highest arsenic toxicity removal rate at the anode is 93.41%. When the electrolyte circulates, the highest arsenic toxicity removal rate is 97.36% at 2.5 cm from the cathode, where the toxic leaching concentration is only 3.40 mg/kg. It can be seen that circulating electrolyte has a greater impact on arsenic leaching toxicity of soil. As shown in Figure 5e, current is an indicator of ion mobility [35]. The influence of circulating electrolyte and non-circulating electrolyte on the operating current is basically the same in the previous period of operation. Both of them increase slowly from the initial current and reach the maximum value at 480 min at the same time, which is 0.072 A and 0.064 A, respectively. Then the current begins to decrease and finally stabilizes. This is because at the initial stage of electrification, there were many electrolytes in the electrode chamber. The concentration of dissolved ions was high, and the current was in direct proportion to the concentration of ions, showing an upward trend. With the repair, the dissolved ions in the soil migrated in the electric field, and the two maintained a dynamic balance, and the current was in a stable state. The concentration of dissolved ions in soil pores decreases with time. It may be that arsenic forms sediment on the surface of PRB and blocks soil pores, and the current intensity is proportional to the concentration of ions in the soil solution. Therefore, the circulating electrolyte has little impact on the residual amount of total arsenic. However, it can change the pH of the bipolar chamber, thus reducing the impact of concentration polarization and improving the toxic removal rate of arsenic in soil.

#### 3.2.2. Electrolyte Type

As shown in Figure 6a, when the electrolyte of the cathode and anode is C_6_H_8_O_7_·H_2_O/KCl, the total arsenic residue at each point is significantly lower, reaching the lowest value of 211.50 mg/kg at the anode, which is also the lowest value of the total arsenic residue in the three groups of soil, indicating that this group has the best arsenic pollution remediation effect. In addition, the total arsenic residue in the soil at the cathode is greater than that at the anode, which shows that the soil remediation effect at the anode is better than that at the cathode, and this phenomenon has not changed due to the difference of electrolyte. As shown in Figure 6b–d, when the cathode and anode electrolyte is KCl/KCl, the anode area is acidic and the cathode area is alkaline, without changing the change rule of pH. In general, during the repair process, the pH value of the soil close to the anode decreases, while the pH value of the soil close to the cathode increases. This is because the H^+^ and OH^−^ produced by the electrolyte migrate to the soil through the applied current, reducing the arsenic electron mobility of the soil, resulting in polarization, and reducing the arsenic removal rate in the soil.

When C_6_H_8_O_7_·H_2_O is used as the cathode electrolyte and KCl is used as the anode electrolyte, due to the buffer capacity of C_6_H_8_O_7_·H_2_O, H^+^ ions can be continuously transported into the soil, while OH^−^ ions are limited in the cathode chamber. This H^+^ ion input effect can reduce the soil pH [36], make the cathode and anode areas acidic, and accelerate the removal and absorption of arsenic by soil particles. C_6_H_8_O_7_·H_2_O is used as the cathode electrolyte and NaOH is used as the anode electrolyte, the OH^−^ ions are confined in the cathode chamber, and the H^+^ in the anode chamber reacts with NaOH, making the cathode and anode areas weakly alkaline. When KCl/KCl is used as the cathode and anode electrolyte, the soil conductivity decreases from the cathode to the lowest value of 401.5 μS/cm at a distance of 2.5 cm from the cathode and then increases all the way to the maximum value of 2831.0 μS/cm at the anode. In the other two groups of experiments using C_6_H_8_O_7_·H_2_O as cathode electrolyte, the cathode began to gradually decrease to the anode, with the highest values of 1951.0 μS/cm and 908.0 μS/cm respectively, and the lowest values of 1379.5 μS/cm and 295.2 μS/cm, respectively.

It can be seen that when C_6_H_8_O_7_·H_2_O is used as the cathode electrolyte, the H^+^ and OH^−^ produced by the electrolyte can be well controlled, thus reducing the electrical conductivity in the soil. The type of electrolyte has a great influence on the removal rate of arsenic toxicity in soil. The removal rate of arsenic toxicity in the soil with KCl/KCl electrolyte as cathode and anode is higher than that in the middle soil. The lowest and highest removal rates of arsenic toxicity are 46.10% and 93.41%, respectively. The removal rate of arsenic toxicity in soil with the cathode and anode electrolyte of C_6_H_8_O_7_·H_2_O/NaOH at each point was lower than that of the other two groups without regularity, and the highest value was 45.34% at the anode. The difference between the two groups is that the removal rate of soil arsenic toxicity changes most regularly when the cathode and anode electrolyte is C_6_H_8_O_7_·H_2_O/KCl. It gradually increases from the cathode to the anode, and finally the maximum value is 92.64% at the anode. At this time, the leaching concentration of soil arsenic toxicity is 11.93 mg/kg. In general, the average removal rate of arsenic toxicity in soil with KCl/KCl as cathode and anode electrolyte is the highest.

As shown in Figure 6e, by changing the type of electrode solution, when the cathode pH is controlled by C_6_H_8_O_7_·H_2_O alone, the current value at the end of operation is 0.141 A, when the anode is controlled by NaOH, and when the cathode is controlled by C_6_H_8_O_7_·H_2_O, the current value at the end of operation is 0.074 A, which is higher than the operating current when the electrolyte is KCl/KCl. This is because the precipitation of other metal ions in the soil decreases and the number of ions increases through the adjustment of the electrolyte pH. As a result, the operating current increases. Therefore, the arsenic residue of the two groups with 0.5 mol/L C_6_H_8_O_7_·H_2_O, 0.1 mol/L KCl and 0.5 mol/L C_6_H_8_O_7_·H_2_O and 0.1 mol/L NaOH electrolyte passing through the cathode and anode after soil remediation is lower than that of the other groups. Their pH value and conductivity are relatively stable without polarization, which can well shield the phenomenon of resistance polarization and concentration polarization during the remediation process.

#### 3.2.3. Anion and Cation Membranes

As shown in Figure 7a, the residual amount of total arsenic in the soil using anion and cation membranes gradually decreases from the cathode to the anode, and its minimum value at the anode is 329.71 mg/kg. The residual amount of total arsenic in the soil using filter cloth fluctuates greatly. The values of cathode and anode are 407.88 mg/kg and 308.67 mg/kg, respectively, which are smaller than those in the soil using anion and cation membranes. Therefore, the use of anion and cation membranes has a greater impact on the residual amount of arsenic in the soil. As shown in Figure 7b–d, the maximum pH values of the soil using filter cloth and anion cation membrane are 9.88 and 9.06, respectively, and the minimum values are 3.28 and 3.27, respectively. The minimum pH values of the soil at the anode are almost the same. It can be seen that the pH values of the soil change gradually from the strong alkalinity of the cathode to the strong acidity of the anode. The change of the soil pH value has little relationship with the use of filter cloth or anion cation membrane.

The difference of soil conductivity at anode is the largest. The soil conductivity using filter cloth is 2831.0 μS/cm. The soil conductivity using anion cation membrane is 1566.0 μS/cm. This is because a cation exchange membrane is placed between the soil and the cathode chamber to prevent hydroxyl ions from entering the soil and depositing with metals as insoluble hydroxide [37]. Only metal cations are allowed to pass through, while other metal ions in the soil are precipitated at the anode. The soil conductivity at the anode is low. An anion exchange membrane is placed between the soil and the anode electrode chamber. Both As^3+^ and As^5+^ in the soil can form oxygen anion protonation in aqueous solution. The main forms of H_2_AsO_4_^−^ and HAsO_4_^2−^ exist [38]. Through the anion exchange membrane, they can be quickly adsorbed at the anode. The conductivity difference between the two groups of soil at the cathode is not significant. The conductivity of soil using filter cloth and anion cation membrane at the cathode is 808.2 μS/cm and 387.5 μS/cm, respectively. The difference at the anode is the largest, the soil conductivity using filter cloth is 2831.0 μS/cm, and the soil conductivity using anion cation membrane is 1566.0 μS/cm.

The change rule of arsenic toxicity removal rate of soil using anion cation membrane and filter cloth is basically the same, which starts from the cathode to decline, begins to rise at 5.0 cm from the cathode to the anode, and obtains the maximum value at the anode. The maximum arsenic toxicity removal rate of soil using filter cloth and anion cation membrane is 93.41% and 89.86%, respectively. It can be seen that the change of arsenic toxicity removal rate of soil has little relationship with the use of filter cloth or anion cation membrane. As shown in Figure 7e, the initial current of the soil using the filter cloth is 0.290 A, which increases to the maximum current value of 0.072 A at 480 min, then gradually decreases with small fluctuations until the end of the operation, and the final current value is 0.046 A. The initial current of soil using anion cation membrane is 0.020 A, the highest value is 0.056 A, and the final current is 0.027 A.

It can be seen that the initial current and termination current of soil using anion cation membrane are smaller than those in the filter cloth group, which may be related to the structure of ion membrane. Because the ion membrane is composed of sulfonic acid and carboxylic acid resin layer, the resistance of carboxylic acid layer is large, which will affect the current. Therefore, the use of anion and cation membranes makes it impossible for the current to carry high concentration ions from one electrode chamber to another, and the current efficiency is greatly improved, resulting in a total arsenic residue at the anode of 329.71 mg/kg, effectively enhancing the effect of electric repair and shielding the concentration polarization phenomenon.

#### 3.2.4. The Exchange of Cathode and Anode

As shown in Figure 8a, the maximum value of total arsenic residue in soil after electrode exchange is 468.13 mg/kg, and the minimum value is 384.61 mg/kg at anode (the cathode after exchange is regarded as anode). At 2.5 cm and 7.5 cm away from the cathode (the exchanged anode is regarded as the cathode), the residual amount of total arsenic in the soil has increased, which is due to the repeated migration of arsenic in the soil caused by the exchange of electrodes [39]. The residual amount of total arsenic in the soil without exchanging the cathode and anode has a large change, the maximum value is 517.40 mg/kg at 2.5 cm away from the cathode, and the minimum value is 313.19 mg/kg at the anode. The residual amount of total arsenic in the soil has little relationship with the exchange of the cathode and anode. As shown in Figure 8b–d, the pH value of the soil without exchanging electrodes decreases gradually from the cathode to the anode, from the highest pH value of the cathode, 9.17, to the lowest pH value of the anode, 2.91, while the pH value of the soil after exchanging the cathode and anode gradually increases from the lowest pH value of the cathode 3.17 to the anode, reaching the highest pH value of 8.50 at the anode. It can be seen that the exchange of the cathode and anode leads to the change of the reaction between the cathode and anode, which has a great impact on the pH value of the two poles of the soil, but the change range of the pH is reduced. The anions and cations generated by the electrolytic reaction are partially neutralized by the newly generated ions after the exchange of the electrode [40], and the soil acidification and soil alkalization near the electrode are alleviated.

The conductivity of the soil with unexchanged cathode and anode electrodes is from 1262.5 μS/cm at the cathode, began to decline, and dropped to the lowest value 634.8 μS/cm at 2.5 cm from the cathode, then rose all the way up to a maximum of 2457 μS/cm at the anode. However, the soil conductivity of the exchanged cathode and anode electrode decreases gradually from the cathode, and the maximum conductivity at the cathode is 2430 μS/cm down to the lowest value 826 μS/cm at 7.5 cm from the cathode, the conductivity of soil at the last anode is 1004 μS/cm. It can be seen that due to the exchange of the cathode and cathode, the conductivity of the anode that should have been higher than the cathode is lower than the conductivity of the cathode after the replacement of the electrode, which may be because the change of the electrode affects the concentration of H^+^ in the soil at the two poles.

After electrode exchange, the highest arsenic toxicity removal rate was 96.56% at the cathode. At this time, the soil arsenic toxicity leaching concentration was 3.30 mg/kg. The highest arsenic toxicity removal rate was 93.41% when the electrode was still at the anode. At this time, the toxic leaching concentration was 10.68 mg/kg. Electrode exchange improved the removal rate of soil arsenic toxicity. As shown in Figure 8e, with the repair process, the current shows different trends due to the exchange of cathode and anode. Compared with the non-exchangeable anode and cathode electrodes, the current increased significantly after about 3000 min. More and more exchangeable metal ions were desorbed from the soil to the soil pore fluid, which promoted the increase of current. Therefore, the exchange of cathode and anode can neutralize the H^+^ and OH^−^ produced by the electrolyte, effectively reducing the accumulation of soil cathode area, shielding the reduction of repair efficiency caused by resistance polarization, enhancing the current, and improving the removal rate of arsenic in soil.

## 4. Summary and Concluding Remarks

In this study, soil arsenic remediation efficiency in EK-PRB was improved by phosphate pretreatment of soil and circulation of electrolyte, C_6_H_8_O_7_·H_2_O as cathodic electrolyte, anionic and cationic membranes, and electrode interchange to produce a shielding effect on the polarization phenomenon. Thus, the EK remediation combined with LDH PRB has a huge potential for As-contaminated soil remediation.

By using self-made Fe/Mn/C-LDH materials as PRB filler, the effects of pretreatment and polarization shielding on EK-PRB of Fe/Mn/C-LDH for remediation of arsenic contaminated soils were investigated. For the pretreatment, phosphoric acid, phosphoric acid and water washing, and phosphate were adopted to reduce the influence of iron in soil. The addition of phosphate could effectively reduce the soil leaching toxicity concentration. The removal rate of the soil pretreated with phosphoric acid or phosphoric acid and water washing was better than with phosphate pretreatment. However, the addition of phosphoric acid changed the original pH value and physicochemical properties of soil. Therefore, the pretreatment method of phosphate was more consistent with the actual project.

For the polarization shielding, the electrolyte circulates from the cathode chamber to the anode chamber through the peristaltic pump to control the pH value of the electrolyte, and the highest arsenic toxicity removal rate in the soil reaches 97.36%. C_6_H_8_O_7_·H_2_O is used as the cathode electrolyte to neutralize the OH^−^ produced on the cathode, maintain the pH value, and prevent the formation of other insoluble salts near the cathode. The variation of total arsenic residue in soil using anion and cation membranes is the most regular. The total arsenic residue gradually decreases from cathode to anode. Electrode exchange can neutralize H^+^ and OH^−^ produced by electrolyte, reduce the accumulation of soil cathode area, shield the reduction of repair efficiency caused by resistance polarization, enhance current, and improve the removal rate of arsenic in soil.

The EK-PRB technology combines the advantages of electric remediation technology and osmotic reaction wall technology. The development of electrokinetic remediation has opened up a new situation for the remediation of contaminated soil. The development of enhanced technology overcomes the limitations of different pollutants and soils on electrokinetic remediation and expands the application of electrokinetic technology in soil remediation. This article provides a comprehensive and updated list of references for readers interested in specific enhancement technologies for further research.

## Figures and Tables

**Figure 1 nanomaterials-13-00325-f001:**
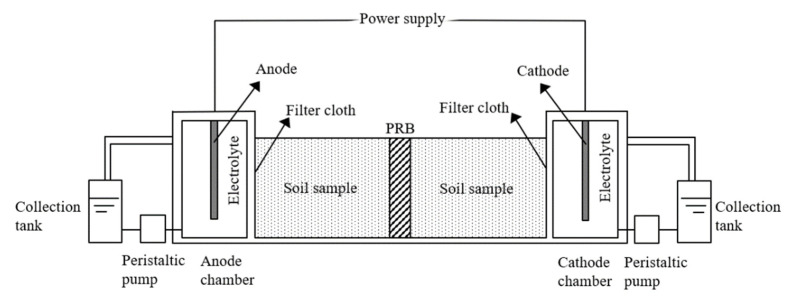
Schematic diagram of EK-PRB reaction device.

**Figure 2 nanomaterials-13-00325-f002:**
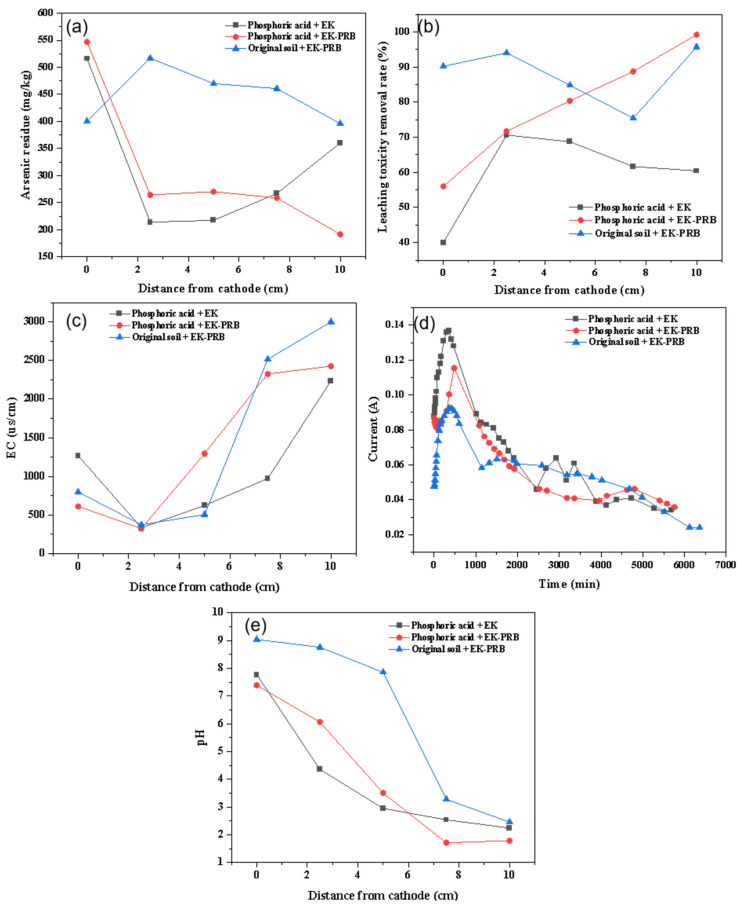
Effect of phosphoric acid for the remediation of arsenic contaminated soil by EK-PRB. Arsenic residue (**a**), leaching toxicity removal rate (**b**), EC (**c**), current (**d**) and pH (**e**).

**Figure 3 nanomaterials-13-00325-f003:**
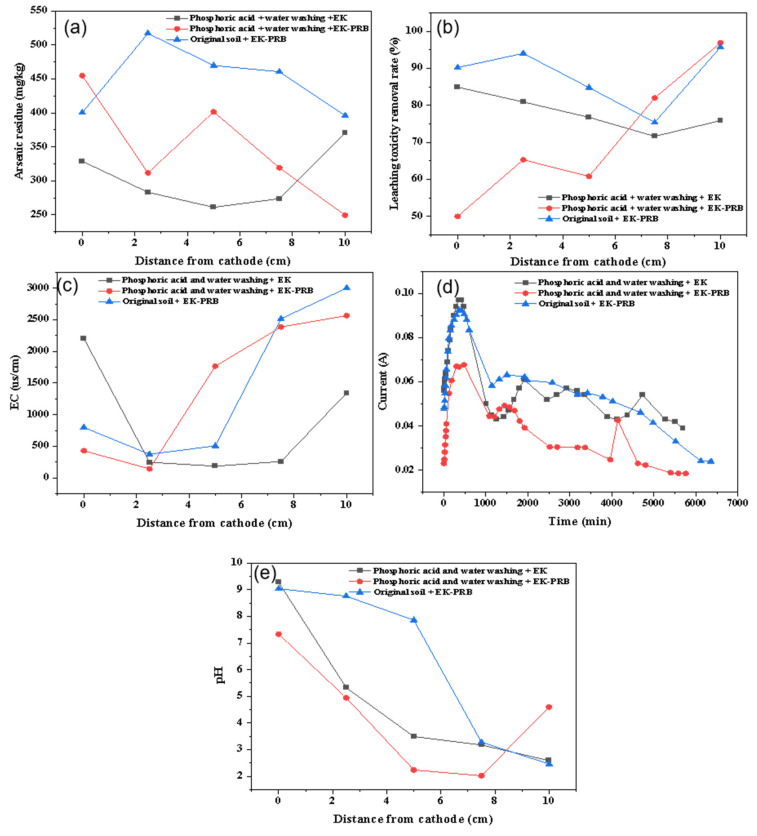
Effect of phosphoric acid and water washing on EK-PRB remediation of arsenic-contaminated soil. Arsenic residue (**a**), leaching toxicity removal rate (**b**), EC (**c**), current (**d**) and pH (**e**).

**Figure 4 nanomaterials-13-00325-f004:**
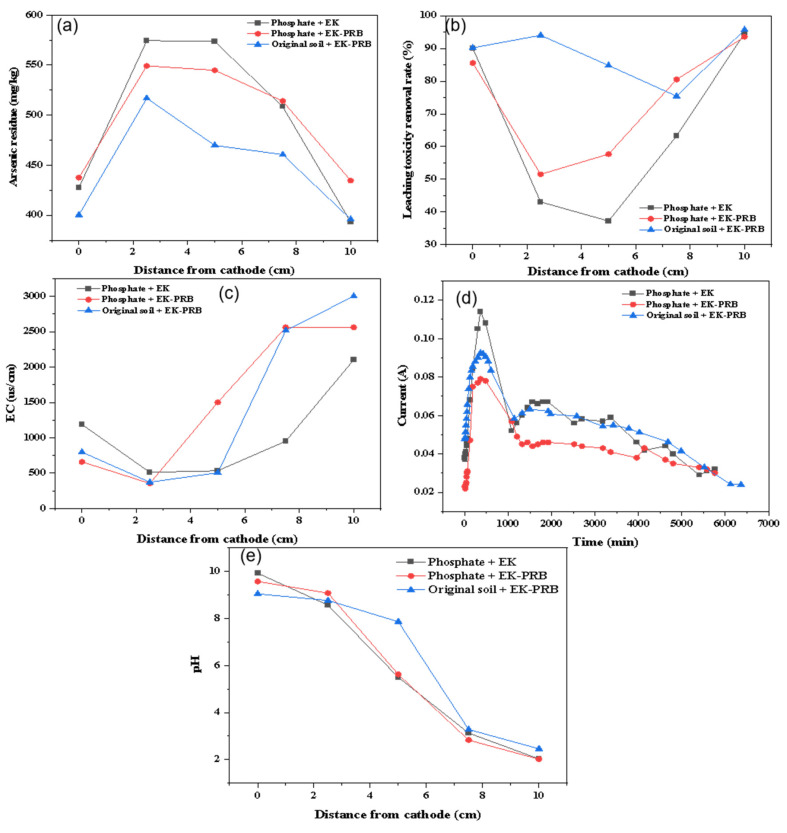
Effect of phosphate for the remediation of arsenic contaminated soil by EK-PRB. Arsenic residue (**a**), leaching toxicity removal rate (**b**), EC (**c**), current (**d**) and pH (**e**).

**Figure 5 nanomaterials-13-00325-f005:**
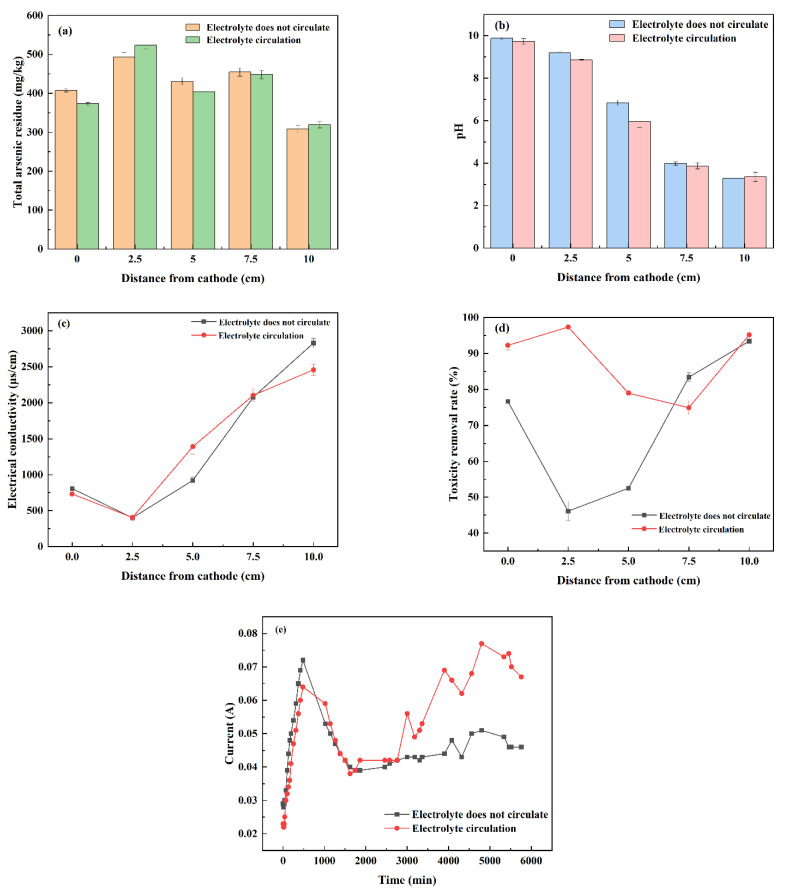
Effect of circulating/non-circulating electrolyte on polarization phenomenon for the remediation of arsenic-contaminated soil by EK-PRB. Total arsenic residue (**a**), pH (**b**), electrical conductivity (**c**), toxicity removal rate (**d**) and current (**e**).

**Figure 6 nanomaterials-13-00325-f006:**
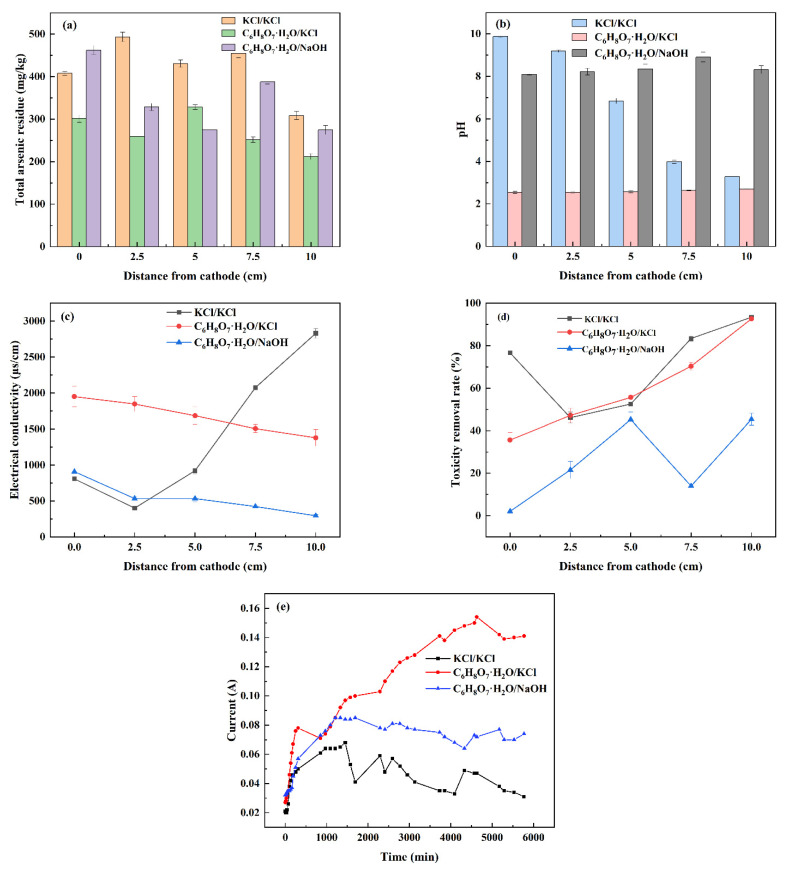
Effect of electrolyte type on polarization phenomenon for the remediation of arsenic contaminated soil by EK-PRB. Total arsenic residue (**a**), pH (**b**), electrical conductivity (**c**), toxicity removal rate (**d**) and current (**e**).

**Figure 7 nanomaterials-13-00325-f007:**
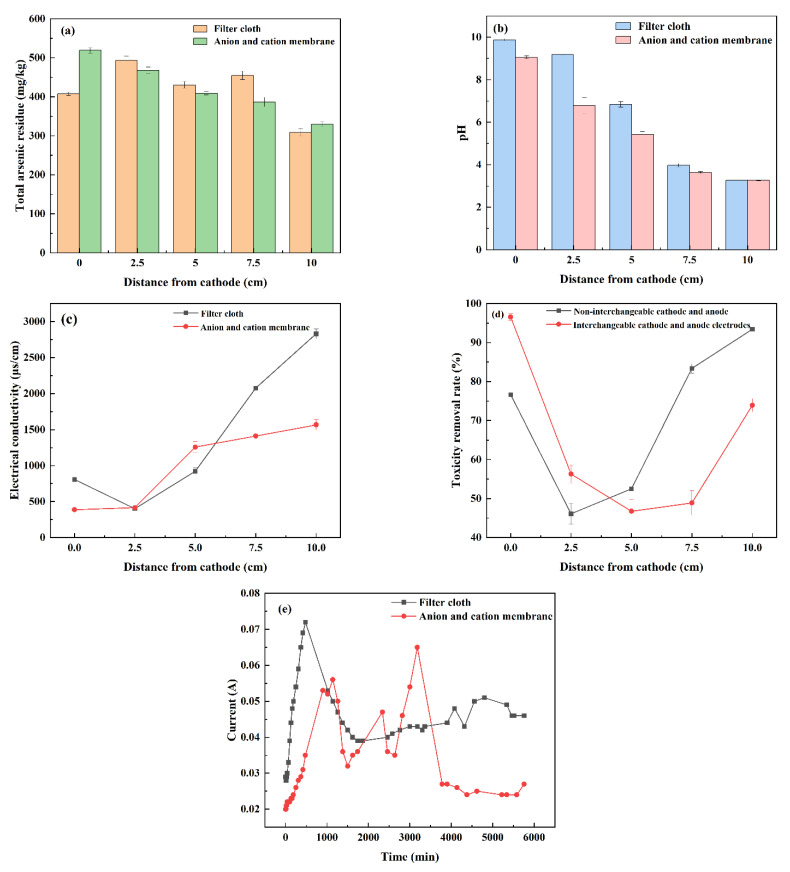
Effect of anion and cation membranes on polarization phenomenon for the remediation of arsenic-contaminated soil by EK-PRB. Total arsenic residue (**a**), pH (**b**), electrical conductivity (**c**), toxicity removal rate (**d**) and current (**e**).

**Figure 8 nanomaterials-13-00325-f008:**
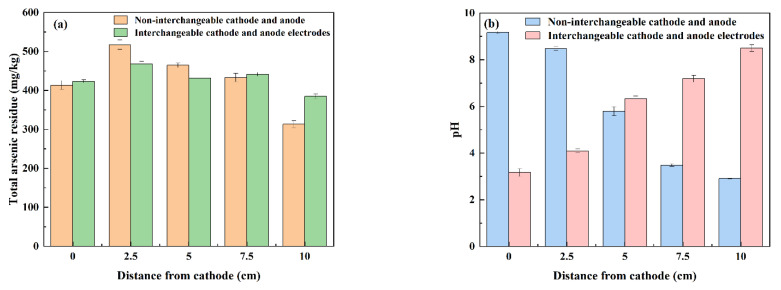
Effect of the exchange of cathode and anode on polarization phenomenon for the remediation of arsenic-contaminated soil by EK-PRB. Total arsenic residue (**a**), pH (**b**), electrical conductivity (**c**), toxicity removal rate (**d**) and current (**e**).

## Data Availability

Not applicable.

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
