# Peer review of "Effects of Pretreatment and Polarization Shielding on EK-PRB of Fe/Mn/C-LDH for Remediation of Arsenic Contaminated Soils"

_nanomaterials, 2023, doi:10.3390/nano13020325_

Round 1

Reviewer 1 Report

Comments

1.      The author please check line number 88. Some words underlining.

2.      Need to improve the figure quality. The author please check all figures.

3.       The author should include the chemical manufacturing, and country name in the materials and methods section.

4.      Conclusion: Strengthen the section by adding novelty, limitations, and implications of the study.

5.      Grammar and typos. The manuscript contains some grammatical and typographical errors. The authors need to thoroughly revise the manuscript and correct the errors.

Author Response

Reviewers' Specific comment

Our response and revision

(Revise according to reviewers’ advices)

The author please check line number 88. Some words underlining.

Thank for this comment. This error is from our carelessness, and it has been corrected, which updated in the revised manuscript. Revise the following sentence in the text:

Due to layered double hydroxides and carbon nanomaterials (C-LDHs) having versatile properties and many composition variables available for tuning, hybrids of them are hopeful nanomaterials. 

Need to improve the figure quality. The author please check all figures.

We appreciate the reviewer very much for the comment. We have revised the problems, which updated in the revised manuscript.

 The author should include the chemical manufacturing, and country name in the materials and methods section.

Revise the following sentence in the text:

Revised manuscript 115-116 lines added: The chemicals were of analytical grade or higher and purchased from the Shanghai Guoyao reagent group, Shanghai, China.

Conclusion: Strengthen the section by adding novelty, limitations, and implications of the study.

Revise the following sentence in the text:

Revised manuscript 513-517 and 539-546 lines added: In this study, Soil arsenic remediation efficiency in EK-PRB was improved by phosphate pretreatment of soil and circulation of electrolyte, C6H8O7·H2O as cathodic electrolyte, anionic and cationic membranes, and electrode interchange to produce shielding effect on polarization phenomenon. Thus, the EK remediation combined with LDH PRB has a huge potential for As-contaminated soil remediation.

The EK-PRB technology combines the advantages of electric remediation technology and osmotic reaction wall technology. The development of electrokinetic remediation has opened up a new situation for the remediation of contaminated soil. The development of enhanced technology overcomes the limitations of different pollutants and soils on electrokinetic remediation, and expands the application of electrokinetic technology in soil remediation. This article provides a comprehensive and updated list of references for readers interested in specific enhancement technologies for further research.

Grammar and typos. The manuscript contains some grammatical and typographical errors. The authors need to thoroughly revise the manuscript and correct the errors.

Revise the following sentence in the text:

Revised manuscript 18 lines: Corresponding authorsï¼›

Revised manuscript 56 lines: electromigration and other mechanismsï¼›

Revised manuscript 58 lines: used zero-valent ironï¼›

Revised manuscript 71 lines: The removal efficiency of the heavy metalï¼›

Revised manuscript 73-77 lines: Taking advantage of the high capacity for anionic exchanging, the LDH is a promising PRB material [15]. It is often applied as adsorption materials, which is known as “anionic clays”, with a wide range of compositions [16]. However, the arsenic was intensively affined to Mn, Fe and Al oxides in soils, leading to low removal rate and poor mobility [17-18].

Revised manuscript 90-91 lines: Due to layered double hydroxides and carbon nanomaterials (C-LDHs) having versatile properties and many composition variables available for tuning;

We proofread the full text, which has been revised and noted in the revised version.

Reviewer 2 Report

Please see detailed comment to authors in the attached page

Author Response

Reviewers' Specific comment

Our response and revision

(Revise according to reviewers’ advices)

Some sentences offered in the introduction should be moved to the conclusions section. One example is the one located between 69-74. Please check for other cases similar to this in the section.

The following sentences are moved to 4 Summary and Concluding Remarks:

The EK-PRB technology combines the advantages of electric remediation technology and osmotic reaction wall technology.

Thus, the EK remediation combined with LDH PRB has a huge potential for As-contaminated soil remediation.

The sketch of the device is mentioned in lines 106-07; however, no detail is available in the draft. I would recommend including a sketch and clearly mark the position of electrodes and other proper geometrical dimensions. Further details could be included in the extra materials. Please refer to this sketch when describing the experimental efforts.

Rewrite manuscript 2.2 Construction and operation of EK-PRB device, add sketches and supplementary material in detail.

A procedure is mentioned in lines115-16, however I would recommend a summary to be included with details given in the extra material section.

Added in Supplementary materials:

We will supplement the detailed steps in the supplementary material 1.1.2Preparation of PRB filler Fe / Mn / C-LDH.

The Engllish should be improved; there are several sentences with verb that should be adjusted.For example, in like 117, “The details were shown…” should read, “The details are included in…”.Another example: “Line 129-130, “it showed that..” should read “It was found that….”.Please check for other similar examples across the draft.

Are there reference materials available? If there are why not use them?

Change the expression:

“The details were shown…” was change to “The details are included in…”

“it showed that..” was change to “It was found that…”

I would recommend the figures that shows less data points see, for example, Fig 1 a,b,c &e) be plotted in a different fashion: For example considering bars for them. Also, I did not see any “errors” bars. It would be advisable to add them or offer an explanation (statistically speaking) why they are not included.

Redrawing the figures and Explanation:

We have redrawn some of the figures. Because phosphate pretreatment soil experiments are stable, there is no error bar in Figures 2 to 4.

I would recommend the results be offered first, then an explanation of why they have a given trend and make concoctions to the application. A typical example is located between lines 129-132 and there several others like this.

Revise the following sentence in the text:

The experiment was carried out in the EK-PRB repair device, which is shown in Figure 1. The EK-PRB remediation device was mainly composed of a polymethyl methacrylate chamber (300 mm×100 mm×50 mm), including electrode, direct current (DC) power supply and PRB filler. The anode and cathode were graphite plates (100 mm×10 mm×50 mm), and the Fe/Mn/C-LDH material was used as the PRB filler. In each run for 96 h, 600 g of the arsenic-contaminated soil were placed into the soil chambers. The voltage, PRB location, moisture content, PRB types and pretreatment of arsenic-contaminated soils were varied to investigate the effects of the operation condition. The detail steps of EK-PRB remediation device are included in the Supplementary material.

The section “Conclusions” should be changed to “Summary and Concluding Remarks”

Revise the following sentence in the text:

4 Summary and Concluding Remarks

I would recommend drafting the last section in the following order: First summarize the results, then connect them to the objectives of the research and, finally indicate what was found and how these results help to make progress or close research gaps in the literature.

Corrected and revise the following sentence in the text:

In this study, Soil arsenic remediation efficiency in EK-PRB was improved by phosphate pretreatment of soil and circulation of electrolyte, C6H8O7·H2O as cathodic electrolyte, anionic and cationic membranes, and electrode interchange to produce shielding effect on polarization phenomenon. Thus, the EK remediation combined with LDH PRB has a huge potential for As-contaminated soil remediation.

The EK-PRB technology combines the advantages of electric remediation technology and osmotic reaction wall technology. The development of electrokinetic remediation has opened up a new situation for the remediation of contaminated soil. The development of enhanced technology overcomes the limitations of different pollutants and soils on electrokinetic remediation, and expands the application of electrokinetic technology in soil remediation. This article provides a comprehensive and updated list of references for readers interested in specific enhancement technologies for further research.
